# Differential Pathogenic Gene Expression of *E. histolytica* in Patients with Different Clinical Forms of Amoebiasis

**DOI:** 10.3390/microorganisms8101556

**Published:** 2020-10-09

**Authors:** Enrique González-Rivas, Miriam Nieves-Ramírez, Ulises Magaña, Patricia Morán, Liliana Rojas-Velázquez, Eric Hernández, Angélica Serrano-Vázquez, Oswaldo Partida, Horacio Pérez-Juárez, Cecilia Ximénez

**Affiliations:** Laboratorio de Inmunología, Unidad de Investigación en Medicina Experimental, Facultad de Medicina, UNAM, México City 04510, Mexico; rivasenrimx@yahoo.com.mx (E.G.-R.); mirnieves@yahoo.com.mx (M.N.-R.); ulisesmagnun@gmail.com (U.M.); patricia_morans@yahoo.com.mx (P.M.); lhily@yahoo.com (L.R.-V.); ericghdz@yahoo.com.mx (E.H.); anseva_31@yahoo.com.mx (A.S.-V.); oswpartida@yahoo.com.mx (O.P.); daztlan13@hotmail.com (H.P.-J.)

**Keywords:** *Entamoeba histolytica*, amebic liver abscess, genital amoebiasis, quantitative polymerase chain reaction, pathogenic genes, differential expression

## Abstract

The etiological agent of human amoebiasis is the protozoan parasite *E. histolytica*; the disease is still an endemic infection in some countries and the outcome of infection in the host infection can range from asymptomatic intestinal infection to intestinal or liver invasive forms of the disease. The invasive character of this parasite is multifactorial and mainly due to the differential expression of multiple pathogenic genes. The aim of the present work was to measure the differential expression of some genes in different specimens of patients with amoebic liver abscess (ALA) and specimens of genital amoebiasis (AG) by RT-qPCR. Results show that the expression of genes is different in both types of samples. Almost all studied genes were over expressed in both sets of patients; however, superoxide dismutase (*Ehsod*), serine threonine isoleucine rich protein (*Ehstirp*), peroxiredoxin (*Ehprd*) and heat shock protein 70 and 90 (*Ehhsp-70*, *EHhsp-90*) were higher in AG biopsies tissue. Furthermore, cysteine proteinases 5 and 2 (*Ehcp*5, *Ehcp*2), lectin (*Ehgal/galnaclectin*) and calreticulin (*Ehcrt)* genes directly associate with pathogenic mechanisms of *E. histolytica* had similar over expression in both AG and ALA samples. In summary the results obtained show that trophozoites can regulate the expression of their genes depending on stimuli or environmental conditions, in order to regulate their pathogenicity and ensure their survival in the host.

## 1. Introduction

*Entamoeba histolytica* (*Eh*) is a protozoan parasite responsible for gastrointestinal amoebiasis in the human host. The host–parasite relationship in this infection can be very diverse and lead to a commensal relationship between the parasite and the host, as in the case of asymptomatic cyst carriers or a relationship in which the parasite causes different degrees of tissue damage that can lead to the invasion of extra intestinal organs [1,2,3].

*E. histolytica* infection is mainly acquired by accidental ingestion of mature cysts present in contaminated water and food (intake of raw and poorly washed vegetables or fruits). It can be transmitted from person to person via the fecal–oral route, contact with hands, objects or surfaces contaminated with infected feces, the cysts of the infected person can remain viable for about 10 to 45 min under the nails [2,3,4,5]; in some cases, it can also be sexually transmitted [6,7].

Amoebiasis is usually a relatively frequent intestinal infection in young adults, in most cases it is asymptomatic because trophozoites remain confined to the intestinal lumen (non-invasive luminal amoebiasis). In the event of symptoms, these appear after an incubation period of two to four weeks and are due to trophozoites invading the intestinal mucosa (invasive form) giving rise to the well-known acute invasive colitis or amoebic dysentery, which consists of diarrhea with mucus and/or blood.

In some patients, such as infants, fever may occur [2]. Amoebic colitis in adults is frequently self-limiting and symptoms disappear within 4 to 7 days, certain individuals may develop chronic luminal amoebiasis that persists for a few months and resolves spontaneously. However, in other patients, the infection may progress to invasive colitis with the appearance of ulcers in the intestinal mucosa that may involve the *muscularis* mucosa and puncture the intestine, the invasion of deep layers of intestinal tissue can lead to the export of amoebic trophozoites to Portal circulation and their implantation in liver tissue with the consequent development of one or multiple amoebic liver abscesses. Very rarely, the brain, pleura, pericardium, genital-urinary tract or skin are also invaded, symptoms vary depending on the affected organ [2,3,4,5]. In the past century, cases of cutaneous amoebiasis in Mexico were not rare, in particular due to the extension of amoebic liver abscess to the abdominal wall, or the presence of intestinal fistulas in cases of intestinal invasive amoebiasis that can reach the abdominal wall [6]. Genital amoebiasis was frequently observed in infants using diapers with dysentery, causing serious damages to the genital organs [7]. More recently the genital cutaneous amoebiasis is detected as a result of unsafe sexual practice, initial lesions might be red and round that rapidly grow giving rise to an ulcer with red gross edges. These lesions evolve rapidly causing inflammation, necrosis and serious damages to the structure of the genital area [8,9].

Multiple mechanisms contribute to the ability of *E*. *histolytica* to destroy the host intestinal mucosa and cause disease [10]. Following excystation within the small intestinal lumen, trophozoites attach to host mucous and epithelial cells in the colon, largely through the multisubunit amebic GalNAc lectin [11,12]. Trophozoites also secrete multiple cysteine proteases, which degrade mucin and the extracellular matrix [13,14,15], they kill resident host cells through a contact-dependent process that remains poorly understood [16,17,18]. Host cells appear to suffer from a disruption of the cell membrane with rapid changes in intracellular calcium levels [19]. This leads to changes in the host cell that closely resemble apoptosis with membrane blebbing, DNA digestion, and activation of caspase [18,20,21]. Finally, *E. histolytica* trophozoites phagocytize red blood cells and nucleated host cells, the ability to phagocytize host cells are strongly associated with virulence [20,21,22,23].

In the present work we analyze the changes in the expression of pathogenicity associated genes of *E*. *histolytica*; cysteine proteinases 5 and 2 (*Ehcp*5, *Ehcp*2), lectin (*Ehgal/galnaclectin*), calreticulin (*Ehcrt)*, superoxide dismutase (*Ehsod*), serine threonine isoleucine rich protein (*Ehstirp*), peroxiredoxin (*Ehprd*), heat shock protein 70 and 90 (*Ehhsp-70*, *EHhsp-90*) in genital cutaneous amoebiasis and amoebic liver abscess human specimens, taking into account that the type of lesions are different both in their tissue structure and in regeneration capacity.

## 2. Materials and Methods

### 2.1. Obtention of Samples

The present work was designed according to the guidelines for the management of human samples for experimental purposes as indicated in the Official Regulation NOM-12SSA3-2007 included in the General Health Law of Mexican Health Ministry. In addition, the project was approved by the Scientific and Ethics Committee of Faculty of Medicine from National Autonomous University of Mexico (project approval number 091/2013).

Samples were obtained from patients admitted in the services of Internal Medicine, Gastroenterology and Infectious Diseases, at the General Hospital of Mexico in Mexico City. The samples then taken for detection of the parasite by serology or PCR at the Immunology laboratory in the Unit of Research in Experimental Medicine, Faculty of Medicine, UNAM.

Patients with clinical diagnosis of amoebic liver abscess (ALA) or genital cutaneous amoebiasis (GA) were included in the study after the signing of a letter of informed consent. Samples of feces and blood were obtained from the two groups of patients; in cases of amoebic liver abscess, a sample of abscess drainage material was also included; in genital cutaneous amoebiasis patients, a biopsy of the lesion was also studied.

The presence of *E. histolytica* was directly determined by microscopic analysis in the feces, imprint of damage tissue and molecularly in the biopsies specimens or liver abscess material using PCR technique. In both groups, serological analysis for anti-amoebic antibodies detection were performed. A total of 15 patients with amoebic liver abscess diagnosis and 5 with genital cutaneous amoebiasis were studied.

### 2.2. Microscopic Detection of Entamoeba Trophozoites

Biopsy specimens from patients with genital cutaneous amoebiasis were obtained from ulcer borders located on the penis and scrotum and fixed in (10%) paraformaldehyde for 24 h. The tissue was then segmented, dehydrated, and embedded in paraffin. The tissues (4–5 µm) were cut with a microtome Leica RM 2145 (Buffalo Grove, IL), mounted on slides, and stained with periodic acid-Schiff (PAS) technique [24]. The slides were observed at 10+ and 40+ magnifications with a light microscope. In case of amoebic abscess material, a smear was performed directly over the slide for staining with PAS.

### 2.3. Anti-Amebic Antibody Detection

Anti-amebic antibody detection was performed in serum samples following the ELISA technique as previously described [25]. For the detection of anti-amebic IgG Serum, samples were diluted 1:1000. The antigen used in ELISA was a membrane-enriched extract (1 µg/well) obtained from axenically cultured *E. histolytica* HM1:IMSS as previously described [25]. Afterwards, 50 µL/well of serum dilution was added to each microtiter plate, blocking previously the free antigen spaces with 3% PBS-BSA and subsequently incubated for 2 h at room temperature. Plates were then washed once with PBS-BSA, added to 0.5% Tween-20 (Sigma Chemical Co.) (PBS-BSA-Tw), and twice with PBS-Tw. Anti-human IgG heavy chain-specific antibodies coupled with peroxidase (Zymed Laboratories, San Francisco, CA) were used for the detection of the antigen-antibody reaction (50 µL/well of 1: 1000 dilution). The antibodies were incubated overnight at 4 °C with mildly shaken. After three consecutive washes (3 min each) with PBS-BSA-Tw, 50 µL/well of substrate solution was added [0.1 M citrate buffer pH 4.5 (10 mL) added with 10 mg o-phenylenediamine and 4 µL of 30% H_2_O_2_ solution and allowed to react for 3 min; then, the reaction was stopped with 1 M H_2_S0_4_ (200 µL/well). Plates were read at 490 nm in a Micro-ELISA reader. An ELISA test for the presence of anti-amebic antibodies was considered positive when the O.D. value was above the cutoff point defined as the mean of the respective O.D. values obtained from non-parasitized control individuals plus two standard deviations (0.525).

### 2.4. Obtention of DNA and RNA from Biological Samples

The QIAamp DNA Mini Kit (QIAGEN, Valencia, CA, USA) was used to extract DNA, following the manufacturer’s instructions, from the necrotic and bloody secretion specimens taken from amoebic liver abscesses specimens, or the biopsy fragment of the lesion in genital cutaneous amoebiasis.

Total RNA was extracted from each of the samples (biopsy fragments, abscess material) by Trizol reagent method (Invitrogen; Life Technologies) and the yield and purity determined by the ratio of absorbance at 260/280 nm (NanoDrop, Thermo Scientific). and DNA was digested using DNAse1 (Qiagen) according to the manufacturer’s instructions.

### 2.5. cDNA Synthesis

cDNA synthesis was performed using Super Script III Reverse Transcriptase system (Thermo Fisher Scientific) following the indications of the supplier. Briefly, 25 µL DPC, treated water containing 2.5 µL of oligo (dT), 10 mM dNTP mix, 5 µL 5× First-Strand buffer, 1.2 µL 0.1 M DTT, 0.25 µL of recombinant ribonuclease inhibitor (40 U/µL), and reverse transcriptase (100 U/sample) were added whit 1 µg of RNA previously obtained and were incubated at 42 °C for 2 h in a moist environment. Thereafter, synthesized cDNA was recovered and quantified by spectrophotometry at 260/280 nm, to await use in the qPCR assays.

### 2.6. PCR Identification of E. histolytica

DNA was amplified using specie specific oligonucleotides; Psp5 GGCCAATTCATTCAATGAATTGAG and Psp3 GGGATCCTGATCCTTCCGCAGGTTCACCTAC previously described by Clark and Diamond (1992) [26]. Both oligonucleotides amplified a product of 876 bp. The PCR reaction was performed adding 1 μL of genomic DNA to the reaction mix: 2 μL PCR buffer consisting of Tris-HCl 10 mM pH 8.3, 0.8 μL de MgCl_2_ (50 μM), 1.6 μL of dNTPs (0.2 mM), 1 μL of ach oligonucleotides (20 μM) and 12.5 µL distillate water. The reaction mix was added with 0.1 μL (0.025 U/μL) of Taq DNA polymerase (AmpliTaq platinum Polymerase, Invitrogen). Amplification was performed in a thermocycler MyCycler Thermal Cycler (BIORAD). Amplification conditions were: DNA denaturation during 5 min at 95 °C followed by 45 cycles of denaturation during 30 sec at 94 °C, 30 sec for alignment at 55 °C; extension takes 1 min at 72 °C and on step of final extension of 10 min at 72 °C; the final volume of reaction was 20 µL.

The PCR products was observed in an electrophoresis in 1% agarose gel. The products were stained with ethidium bromide in a UV chamber. A reference of 1000 bp size and *E. histolytica* DNA control were included.

### 2.7. Molecular Analysis of the Expression of Amoebic Genes

To perform studies of quantification and expression of amoebic genes in the two groups of samples studied, we use Reverse Transcriptase quantitative PCR (RT-qPCR). The specific oligonucleotides for the genes studied are shown in Table 1 (*Ehcp*5, -*cp*2, -*gal/galnaclectin*, -*crt*, -*stirp*, -*prd*, -*sod*, -*hsp*70 y 90). The cDNAs previously obtained, were used as temperate at (15 ng/µL) in the reactions of qPCR using the Step One-Applied Biosystems and using for the reaction the kit (Quantitec SYBER Green PCR, Qiagen). The amplification was developed with 60 cycles with 3 stages; including a denaturing stage at 95 °C for 10 sec, a phase of alignment to 57 °C for 30 sec and a phase of elongation at 72 °C for 10 sec and finally to 4 °C to finish the reaction. The amplification of each gene was tested by tripled and the differential expression of the genes investigated was calculated by normalized to a reference gene (often referred to as the housekeeping gene) the Eh-α-actin of the trophozoites. The data were analyzed using the method of 2^−ΔΔCT^ described by Schmittgen and Livak (2008) [27] and the validation method reported by Yalcin (2004) [28]. Comparing the differential expression of *E. histolytica* genes in the samples from both group of patients.

### 2.8. Statistical Analysis

Results are expressed as the average and the standard deviation of at least two duplicates of each condition and the analysis of independent samples. The statistical analysis was made using the program GraphPad Prism 5 software. We used the Student *t* tests to determine the statistically significance of differences between the studied patients’ samples. The statistical values were considered significant when the *p* value was less than *p* < 0.05.

## 3. Results

### 3.1. General Characteristics of Studied Patients

Fifteen patients with amoebic liver abscess were included in the study, 2 females and 13 males between 25 and 60 years old. All patients were submitted to ultrasound guided drainage of the abscess as indicated by the physician in charge. All patients had high levels of serum anti-amoebic antibodies in the ELISA test (OD = 1.171 ± 0.366) (Figure 1 and Appendix A). The drainage material from the liver abscess was processed for DNA extraction and submitted thereafter to PCR for detection of *E. histolytica* amplification products, all the abscesses specimens show the presence *E. histolytica* PCR amplification products (Figure 2).

The five patients with genital cutaneous amoebiasis were all males between 35 to 55 years old. In each case we were able to detect microscopically the presence of trophozoites of *Entamoeba* (Figure 3). The ELISA test for detection of serum anti-amoebic antibodies was positive in the five cases (OD = 0.833 ± 0.225) (Figure 1), and positive for detection of *E. histolytica* PCR amplification products (Figure 2).

### 3.2. Analysis of the Expression of Amoebic Genes Associated with Pathogenicity

The qPCR technique was used to determine the differential expression of *E. histolytica* genes in the two different forms of invasive amoebiasis studied (amoebic liver abscess and genital cutaneous amoebiasis).

The results obtained show a clear overexpression in all the genes studied (Figure 4). However, this over-expression was higher for genes that encode proteins associated with the protection of trophozoites against stressful environmental conditions, such as *Eh*HSP-70, *Eh*HSP-90, *Eh*PRD, and *Eh*SOD in samples obtained from patients with genital cutaneous amoebiasis (*p* = 0.0027, 0.0013, 0.043, 0.0032). With regard to genes directly associated with pathogenicity factors (*Eh*CP-2 and *Eh*CP-5), a slightly higher overexpression was also observed when compared to patients with amoebic liver abscess but there is no significant difference when compared with the results of the trophozoites present in patients with genital cutaneous amoebiasis (*p* = 0.072, 0.36, respectively) (Figure 4). While for the *Eh*CRT and *Eh*Gal/Galnaclectin genes, the overexpression is mostly the same for both amoebic liver abscess and genital cutaneous amoebiasis infection conditions (*p* = 0.24, 0.092, respectively).

However, in the case of *Eh*STIRP the mayor overexpression was observed in the cases of genital cutaneous amoebiasis (*p* = 0.0038).

## 4. Discussion

During its life cycle, *E. histolytica* is challenged by a wide variety of environmental stresses, such as fluctuation in the concentration of oxygen, glucose, changes in the composition of the intestinal microbiota and the release of reactive oxygen species, and oxidative nitrosatives of neutrophils and macrophages as products of the host’s immune response, so that the survivability of this parasite is continuously tested to adapt and survive the dynamic environment of the host [29,30].

The studies of these adaptive mechanisms with which this parasite has been transformed and facilitated due to the development of genomics, proteomics or metabolomics (OMICS sciences) [31,32]. Many transcriptome-level studies in *E. histolytica* have investigated gene expression patterns to help understand the pathology and biology of the organism. Virulent and avirulent strains have been compared in laboratory culture and after tissue invasion, cells cultured under different stress conditions, and response to anti-amoebic drugs, in order to identify and understand changes in gene expression during the different scenarios [33,34,35,36,37,38].

In this work, we compared the expression level of preselected genes (*Ehcp*5, *Ehcp*2, *Ehgal/galnac lectin*, *Ehcrt*, *Ehstirp*, *Ehprd*, *Ehsod*, *Ehhsp*-70 and 90) of *E. histolytica*, in two of its clinical manifestations the amoebic liver abscess and the genital cutaneous amoebiasis using the qPCR technique, this technique is used as a validation test for gene expression [39,40].

Given the locations of each morphological invasive process, trophozoites will be exposed to physiological and different environmental conditions. The parasite must be able to adapt to the demands of the environment to survive.

In some cases, and after the invading intestinal infection, the trophozoites can reach the bloodstream reaching the Porta circulation travelling to the liver, where it gives rise to one or multiple abscesses. Circulating trophozoites are confronted with the host’s innate immune response, with cytotoxic compounds, as well as with oxygen tensions higher than those usually found in the gut. In addition, trophozoites are exposed to other host defense mechanisms they are notably attacked by reactive oxygen and nitrogen intermediates produced by phagocyte cells, these compounds are highly concentrated in the infected tissue. For example, it has been shown experimentally that trophozoites invading the liver are highly sensitive to blood complement and tissue responses [2,7,41,42].

The invasion in genital cutaneous amoebiasis occurs by contiguity trophozoites from the intestine to the perianal skin, or skin that is around fistulas, colostomy incisions and abscesses drain incisions. This causes epithelial ulcers in this environment, trophozoites are exposed to a higher concentration of oxygen than in the liver [7,41].

In this sense, our results corroborate these findings since the expression of the genes (*Eh-stirp*, -*prd*, -*sod*) underscores the importance of these as they confer the ability of trophozoites to adapt to environments with a higher concentration of oxygen.

Peroxiredoxins of *Entamoeba* have been shown to play a crucial role for protection against oxidative stress, as in virulence, by helping the parasite to survive host immune response. The decreased gene expression causes the accumulation of ROS, a decrease in parasite viability, as well as in its cytotoxicity [43,44,45]. Moreover, the differential expression and immunolocalization of antioxidant enzymes in *E. histolytica* isolates during metronidazole stress have been reported [37].

This adaptive parasite response provides protection against the host response, as well as aiding in its survival. In eukaryotic cells, the overall stress is a perfectly orchestrated response mechanism. The first step is the function of a stress protein sensor (heat shock proteins, nutrient detection and antioxidant proteins as well as chromatin-associated proteins) to transmit the message to the cells and adapt to stress [46,47,48]. This coincides with the overexpression of *Ehhsp*-70 and-90, *Ehprd*, Ehstirp and *Ehsod* in patients GA genes that are exposed to greater thermal stress due to lower skin temperature (<37 °C) and higher O_2_ tension than ALA patients.

Matthiensen et al. (2013), investigate the roles of different papain-like cysteine peptidases (CPs) as pathogenicity factors; they show that the expression of some of the peptidases that are normally expressed at low levels, however, increases during amoebic abscess formation [49].

In this sense, the results obtained in this work show that these cysteine proteinases (*Ehcp5*, -*cp2*) are overexpressed in the same way in the two forms of amoebiasis studied, amoebic liver abscess and genital cutaneous amoebiasis. The findings reinforce the importance of CPs as pathogenicity factors of *E. histolytica*.

Another gene highly expressed in this work, was, *Ehstirp* (one of a family of *E. histolytica* serine-threonine- and isoleucine-rich proteins). There is very little information on the importance of this gene; however, it was identified during the process of adherence and then has been described by transcriptomic studies as a virulence factor [33,50,51,52], apparently absent in Rahman [51]. Analyzing the functional role of this protein with the overexpression reported in this work supporting the hypothesis that *Ehstirp* has a role in amoebic virulence.

It is clear that the expression of these genes (Eh-stirp, prd, sod, Ehhsp-70 and Ehhsp-90) can be regulated by stressful and environmental conditions where the invading process occurs, this environment will be determined not only by the organ invaded, but also by the host’s immune response. As has been observed in other genes, such as the case of AIG1, whose expression contrasts the virulence level of the parasite depending on the genetic background of the parasite, but also on the environmental conditions of the host [53].

The diversity of the compartments that *E. histolytica* crosses during infection, combined with the variability of induced symptoms, suggest that this parasite expresses different transcriptional programs, although the key components of the regulation of gene expression remain undescribed.

The results obtained in this work, could be the beginning of future investigations that shed light on the participation of environmental conditions in the expression of virulence mechanisms in host pathogenic microorganisms.

## Figures and Tables

**Figure 1 microorganisms-08-01556-f001:**
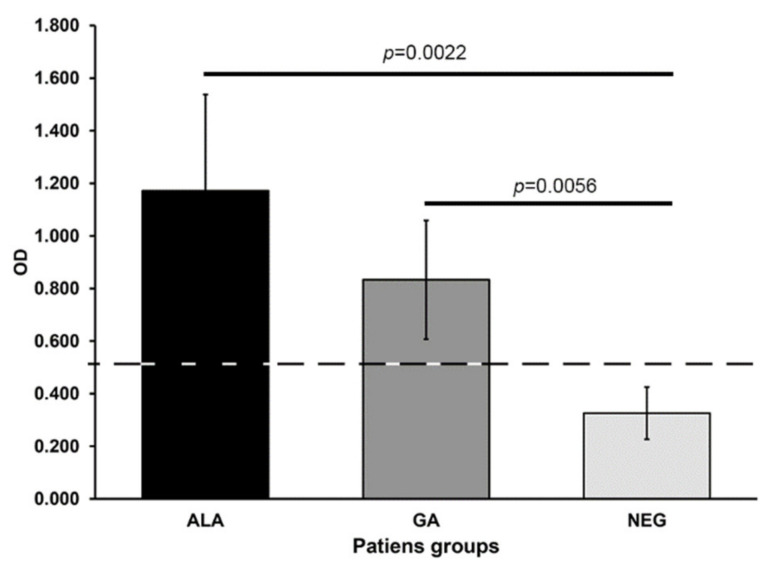
Reactivity of sera from patients with amoebic liver abscess (ALA) and cutaneous genital amoebiasis (GA). The reactivity of human IgG against *E. histolytica* antigen was measured, the average of the OD (590 nm) per group of patients is represented here. Cutline defined as the mean of the respective OD values obtained from non-parasitized individuals (Negative group) plus two standard deviations (0.525) and results previously reported [25]. Statistical significance was included (*p* ≤ 0.05). The individual dates are in the Appendix A.

**Figure 2 microorganisms-08-01556-f002:**
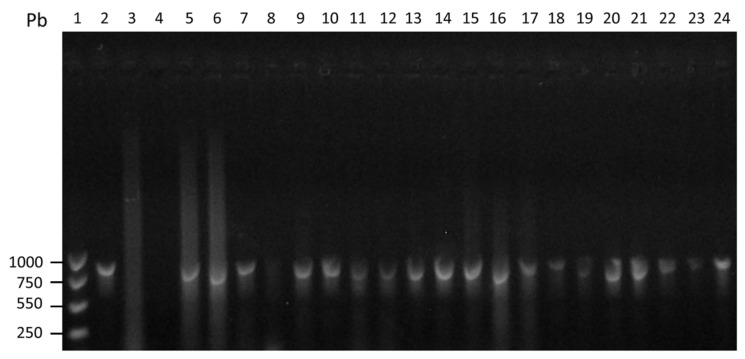
PCR identification of *E. histolytica*. The DNAs obtained from samples of tissue fragments from lesions of genital cutaneous amebiasis patients and abscess drainage from patients with amoebic liver abscess, were evaluated by PCR using oligonucleotides specific (Psp) previously described by Clark and Diamond [26]. Lane (1) 1000 pb mass ruler DNA ladder; (2) Positive control for *E. histolytica* strain HM1: IMSS; (3) Negative control for *E. histolytica*; (4) H_2_0 sample; (5–19) DNA obtained from drainage from patients with amoebic liver abscess; (20–24) DNA obtained from lesions of genital cutaneous amebiasis patients. Products amplified and stained with ethidium bromide, Product size 876 bp.

**Figure 3 microorganisms-08-01556-f003:**
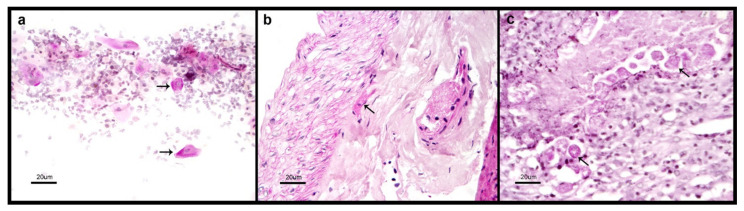
Microscopic identification of *E. histolytica* trophozoites. Biopsy specimens from patients with genital cutaneous amoebiasis histologically treated and stained with PAS, representative images; (**a**) a smear was performed directly over the slide; (**b**,**c**) samples from two different patients. The arrows point out some illustrative trophozoites. The scale bar represents 20 µm.

**Figure 4 microorganisms-08-01556-f004:**
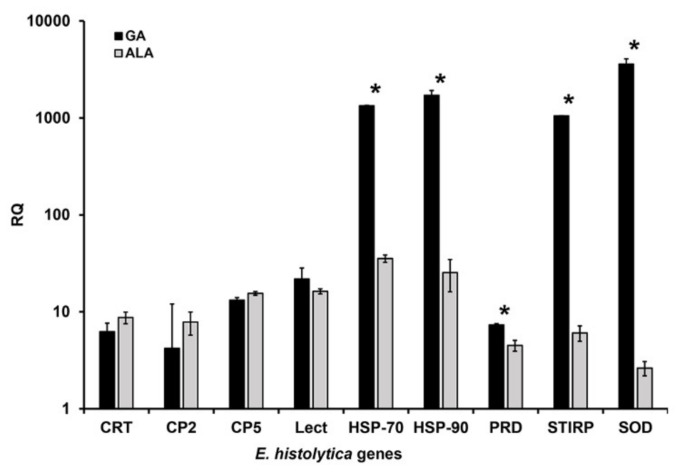
Relative quantification (RQ) of expression of genes of *E. histolytica* by RT-qPCR. Reverse transcription real time PCR was used to independently measure mRNA expression of *Ehcp5, -cp2, -gal/galnac lectin, -crt, -stirp, -prd, -sod, -hsp-70 y 90* genes in trophozoites present in samples of tissue fragments from lesions of genital cutaneous amoebiasis (GA) and abscess drainage from patients with amoebic liver abscess (ALA). The values represent the mean of different patients for group. RQ differences between the cDNAs obtained from samples of tissue fragments from lesions of GA patients and abscess drainage from patients with (ALA) were compared trough Student *t* tests, detecting statistical significance * (*p* ≤ 0.05). *Ehactin*-*α* gen was used as housekeeping gene to normalize mRNA level. The individual dates are shows in the Appendix A.

**Table 1 microorganisms-08-01556-t001:** Oligonucleotides used in the RT-qPCR assay.

Gene	Access Number	Size of Products (pb)	Forward Primer (5′-3′)	Reverse Primer (5′-3′)
*Ehcrt*	XM_650149.1	355	TGGACCAGATGTATGTGGAGG	TGGTGCTTCCCATTCTCCATC
*Ehcp*-5	XM_645845.2	255	GTTGCCGCTGCTATTGATGC	ACCCCAACTGGATAAGCAGC
*Ehcp*-2	XM_645550	115	ATCCAAGCACCAGAATCAGT	TTCCTTCAAGAGCTGCAAGT
*Ehlect*	AF337950.1	281	ACCAGTGAATGGAGCATGTGT	TTG TGC ATT CGC CTT CTC CT
*hprd*	XM_646911.2	121	TCAAGAGAAAGAATGTTGTTGT	ACATGGACAATATGCTGCTGC
*Ehsod*	XM_6437351	172	GCAGCCCAAGCATGGAATCA	ACCAACACCATCCACTTCCA
*Ehstirp*	XM-648869	305	GCTAACAACGCGGAAAGTAGC	ACAAGAGCAGAGCACCCTTC
*Ehhsp*-70	XM_001734367.1	135	GAAACAGAACCACGTCCAGTT	TTACGTCCTCCAAGTCTCCAAT
*Ehsp-90*	AB092411.1	357	ACAAGAGCAGAGCACCCTTC	CCCCATGTGCACTTGTTACAG
*Ehactin*-α	XM_650064.2	211	AGCTGTTCTTTCATTATATGC	TTCTCTTTCAGCAGTAGTGGT

*Ehactin-α* gen of reference. Specificity of the primer pairs for relevant genes was verified by performing nucleotide alignment searches using BLAST.

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
