# Peer review of "Differential Pathogenic Gene Expression of E. histolytica in Patients with Different Clinical Forms of Amoebiasis"

_microorganisms, 2020, doi:10.3390/microorganisms8101556_

Round 1

Reviewer 1 Report

This manuscript describes the differential expression of certain stress-related and virulence-related genes in E. histolytica cells from samples two different clinical manifestations of amoebiasis.

The results are interesting but do not take the story very far. There are also some issues in terms of not showing the data from the individual samples. The antibody reactivity assay suggests different levels of severity of infection and it would be useful to see the results for the individual samples as well as the aggregate. Do samples that have different antibody reactivity show any difference? The small error bars in the qPCR results suggest no. 

Specific suggestions:

  1. show the data for individual samples (perhaps in the supplements) for both the antibody reactivity and qPCR.
  2. Also show the PCR identification of E. histolytica for all of the samples rather than a subset.

The manuscript also needs to be edited for minor language issues but also for general writing issues. Many of the sentences are overly long and difficult to understand (e.g., lines 48-53) and other parts of the manuscript are very choppy and do not flow well. For example, there are multiple paragraphs on the last page of the discussion that are a single sentence.

Author Response

The results are interesting but do not take the story very far. There are also some issues in terms of not showing the data from the individual samples. The antibody reactivity assay suggests different levels of severity of infection and it would be useful to see the results for the individual samples as well as the aggregate. Do samples that have different antibody reactivity show any difference? The small error bars in the qPCR results suggest no. 

Response: We agree with our reviewer, these results could be the beginning of future investigations that shed light on the role of environmental conditions in the expression of E histolytica virulence mechanisms in the human host

Regarding the results of serology, the differences observed in the levels of anti-E. histolytica antibodies between both groups of patients (ALA and GA) are not statistically significant, however, both groups show statistically significant differences with respect to the control group

Specific suggestions:

  1. Show the data for individual samples (perhaps in the supplements) for both the antibody reactivity and qPCR.

Response: We already included the individual files as supplementary information

  1. Also show the PCR identification of E. histolytica for all of the samples rather than a subset.

Response: We already change the Figure including all the samples

3-The manuscript also needs to be edited for minor language issues but also for general writing issues. Many of the sentences are overly long and difficult to understand (e.g., lines 48-53) and other parts of the manuscript are very choppy and do not flow well. For example, there are multiple paragraphs on the last page of the discussion that are a single sentence.

Response: We make all the changes suggested by our reviewer

Reviewer 2 Report

The authors address the question whether Entamoeba histolytica parasites differ in the expression of selected genes depending on the site of infection. Therefore they obtained biopses from liver abscess patients as well as from patients with genital skin amoebiasis. The parasites were identified by PCR and microscopy in the biopsies from both sites, and anti-amoebic antibodies were measured in the patients´ sera. Then, total RNA was extracted from the biopsies followed by RT qPCR for a panel of genes. In the skin biopsies, heat shock protein genes as well as genes coding for oxidative stress response factors were significantly up-regulated.

  This work reports for the first time gene expression in skin amoebiasis and the differences are an interesting finding. There are a number of queries, however, that need to be addressed:

  1. The authors report the approval from the local Ethics Committee, for completeness, they should provide a project/approval number. Normally, liver biopsies are not needed in uncomplicated cases of amoebic liver abscess. Did the patients suffer from more severe forms of liver abscess?

  1. Fig. 1, serology: expectedly, both groups of patients have elevated levels of antiamoebic antibodies. The skin amoebiasis patients appear to have a lower level of antibodies. Was this significant as well?

  1. Selection of genes: there is one significant mistake, the gene with the accession number AF337950.1 does not encode the best-studied lectin heavy chain, but the intermediate chain about which is known the least. Why not take the heavy chain? Also the old Ehsod accession number X70852.1 is no longer valid, please use the currently available number XM_643735. The reference actin gene is called beta in line 166 but alpha in Table 1.

  1. The Ehstirp protein may be associated with virulence, but it is unfortunate that very little is known about its function, it may have been better to check for amoebapore A expression, a well-known virulence factor

Minor queries:

Line 13, “underdeveloped countries“ is an unpleasant term, who wants to be in such a country? Also amoebiasis is highly prevalent in developed countries such as India, South Africa, Mexico, or even in the USA

Line 20, please explain the gene abbreviations

Line 26, Keywords, quantitative PCR

Line 70, 71, phagocytize

Line 102, micro m (lower case)

Lines 142-153 check for typos

Line 165, what do the authors mean by “tempered by tripled“ ?

Line 283, why could there be more heat stress in the skin amoebiasis?

RT qPCR, for the final version, please check the MIQE guidelines (Bustin et al., 2009) for the details in nomenclature and which data to provide.

Author Response

  1. The authors report the approval from the local Ethics Committee, for completeness, they should provide a project/approval number. Normally, liver biopsies are not needed in uncomplicated cases of amoebic liver abscess. Did the patients suffer from more severe forms of liver abscess?

Response; the approval number was included (line 84)

  1. Fig. 1, serology: expectedly, both groups of patients have elevated levels of antiamoebic antibodies. The skin amoebiasis patients appear to have a lower level of antibodies. Was this significant as well?

 Response; differences observed in the levels of anti-E.histolytica antibodies in both groups (ALA and GA) do not show significant differences, only a statistically significant difference is observed between the groups of patients with invasion and the control group. 

  1. Selection of genes: there is one significant mistake, the gene with the accession number AF337950.1 does not encode the best-studied lectin heavy chain, but the intermediate chain about which is known the least. Why not take the heavy chain? Also the old Ehsod accession number X70852.1 is no longer valid, please use the currently available number XM_643735. The reference actin gene is called beta in line 166 but alpha in Table 1.

Response;  The suggested changes were  made , both the accession number of Ehsod and the errors in relation to  actin  gene in page 166 and Table 1

Why not take the heavy chain? Because, the purpose of this work is to measure the expression of the  lectin gene, if the light chains are expressed, the genes for the heavy chains must also be expressed, and we already had these primers.

4-The Ehstirp protein may be associated with virulence, but it is unfortunate that very little is known about its function, it may have been better to check for amoebapore A expression, a well-known virulence factor

4-Response; It is true that there is very little information about this gene, but what does exist, largely indicates the preponderant role that this protein has in the pathogenicity mechanisms of this protozoan. The results shown in this work reinforce the role of this gene in these mechanisms.

Minor queries:

Line 13, “underdeveloped countries“ is an unpleasant term, who wants to be in such a country? Also amoebiasis is highly prevalent in developed countries such as India, South Africa, Mexico, or even in the USA

Response

Line 20, please explain the gene abbreviations (This is already done)

Line 26, Keywords, quantitative PCR (already corrected)

Line 70, 71, phagocytize (Corrected)

Line 102, micro m (lower case) (Already done)

Lines 142-153 check for typos (Now are corrected)

Line 165, what do the authors mean by “tempered by tripled“Experiments were performed in triplicate” we included the correction.

Line 283, why could there be more heat stress in the skin amoebiasis?  Response: One possible explanation is that E. histolytica is subjected to more thermal stress due to lower skin temperature (<37 centigrade) than trophozoites located in liver tissue, in our opinion, this could be a reasonable possibility.

RT qPCR, for the final version, please check the MIQE guidelines (Bustin et al., 2009) for the details in nomenclature and which data to provide.

Response: We make the changes in accordance with Bustin el al. 2009.

Reviewer 3 Report

While intestinal and liver infection with Entamoeba histolytica has been extensively studied, much less is known about genital infection with the parasite. Here the authors collect Entamoeba RNA from 15 liver infections and five genital infections and use qPCR compare the expression of numerous genes associated with pathogenicity including adherence lectins and heat shock proteins. Interestingly, some of these factors are expressed more in Entamoeba from genital infections than from liver abscesses.

These experiments, which have small samples of patients and few genes examined, appear to be a starting point for more in depth investigations.

The authors need not refer to overexpression of genes unless they are comparing liver abscess versus genital lesions.

Author Response

Response: we are agree, these results could be the beginning of future investigations that shed light on the role of environmental conditions in the expression of E histolytica virulence mechanisms in the human host. But also, qPCR is a validation tool for data obtained by transcriptome analysis. Overexpression refers to an increase in the expression of a gene based on normalization with the reference gene. Therefore, we can compare the expression between the two groups of patients with invasive forms (ALA vs GA) and we find significant differences in the expression of the following genes; (EhHSP-70 and 90, Ehprd, Ehstirp, Ehsod)

Round 2

Reviewer 1 Report

The issues with the manuscript have been appropriately addressed.